# Quantitative assessments of retinal macular structure among rural-dwelling older adults in China: a population-based, cross-sectional, optical coherence tomography study

Qinghua Zhang [1,2,3] Cong Zhang,[2] Yongxiang Wang,[1,2,3] Lin Cong,[1,2,3] Keke Liu,[1] Zhe Xu,[2] Chunyan Jiang,[1] Weiyan Zhou,[4,5] Chunxiao Zhang,[4,5] Yi Dong,[1,2,3] Jianli Feng,[6] Chengxuan Qiu,[1,7] YiFeng Du [1,2,3]

**Correspondence to**
Dr Chengxuan Qiu;
chengxuan.qiu@ki.se and
Dr YiFeng Du;
duyifeng2013@163.com

## ABSTRACT

**Objectives** To quantitatively assess and compare retinal macular structures of rural-dwelling older adults in China using two different optical coherence tomography (OCT) scanners and to examine their associations with demographic, lifestyle, clinical and ocular factors.

**Design, setting and participants** This population-based, cross-sectional study included 971 participants (age ≥60 years) derived from the Multimodal Interventions to Delay Dementia and Disability in Rural China study. We collected data on demographics, lifestyle factors, clinical conditions (eg, cardiovascular disease (CVD)) and ocular factors (eg, visual acuity and spherical equivalent). We used two models of spectral-domain OCT to measure macular parameters in nine Early Treatment Diabetic Retinopathy Study subfields. Data were analysed using the multiple general linear models.

**Results** Spectralis OCT demonstrated higher macular thickness but a lower macular volume than Primus 200 OCT (p<0.05). Nasal quadrant of the inner and outer subfields was the thickest, followed by superior quadrant. Adjusting for multiple potential confounding variables, older age was significantly correlated with lower average inner and outer macular thicknesses and overall macular volume. Men had higher macular parameters than women. The presence of CVD was correlated with lower central macular thickness (β=−6.83; 95% CI: −13.08 to −0.58; p=0.032). Middle school or above was associated with higher average inner macular thickness (β=7.85; 95% CI: 1.14 to 14.55; p=0.022) and higher spherical equivalent was correlated with lower average inner macular thickness (β=−1.78; 95% CI: −3.50 to −0.07; p=0.042).

**Conclusions** Macular thickness and volume assessed by Spectralis and Primus 200 OCT scanners differ. Older age and female sex are associated with lower macular thickness and volume. Macular parameters are associated with education, CVD and spherical equivalent.

## STRENGTHS AND LIMITATIONS OF THIS STUDY

⇒ The strengths of our study include the population-based design that engages rural-dwelling older adults in China, the use of spectral-domain optical coherence tomography (OCT) rather than time-domain-OCT scanners, and standardised assessment of systemic and ocular factors.

⇒ The cross-sectional nature of our study does not allow us to make any causal inference for the observed associations between the examined factors and macular parameters.

⇒ The observed cross-sectional associations are subject to selective survival bias that usually leads to underestimation of the true associations.

⇒ Data on ocular factors were available only in a small sample, resulting in limited power to analyse associations of these factors with macular parameters.

**Trial registeration number** MIND-China study (ChiCTR1800017758).

## INTRODUCTION

The retina develops from the diencephalon and is an extension of the central nervous system (CNS) that exhibits typical properties of the brain.[1] In patients with neurodegenerative disorders such as Parkinson's disease and Alzheimer's disease, macular changes may precede onset of symptoms in the CNS diseases.[2–4] Thus, changes of macular structure may be useful markers for early detection of CNS disorders.

Optical coherence tomography (OCT) is a key non-invasive imaging technique that enables clinicians to quantitatively measure macular thickness and volume in different subfields. The imaging speed (40, 000 axial scans per second) and image resolution (axial

resolution up to 5 μm) of OCT are much higher than that of MRI scans (resolution <100–500 μm), while the expenditure is much lower.[5] Therefore, OCT scan may provide a comprehensive understanding of the macular structure that can be implemented in studies of the general population settings.

In 2010, the population-based Handan Eye Study used the time-domain OCT (TD-OCT) to describe normal macular parameters in rural Chinese adults (n=2230, age 30–85 years).[6] However, OCT technology has advanced remarkably from TD-OCT to spectral-domain OCT (SD-OCT), which may provide more precise segmentation and higher resolution and measurement of retina.[7] Moreover, very few population-based studies have assessed the differences and agreements of macular measurements across various models of SD-OCT scanners. In addition, differences in macular parameters by demographic factors (eg, age and sex) are well known, but ocular and vascular risk factors in association with macular measurements in older adults have not yet been well studied.[8–11]

In this population-based study among rural-dwelling older adults in China, we sought to describe the methodology of quantitative assessments of the macular parameters assessed with SD-OCT scanners, to compare macular parameters measured using two different SD-OCT scanners, and further to examine the relations of macular parameters with demographic features, ocular factors and vascular risk factors. We supposed that macular measurements assessed with different models of SD-OCT scanner differed and we further hypothesised that macular parameters (eg, macular thicknesses and volume) in older adults were associated with demographic, lifestyle, clinical and ocular factors.

## METHODS
### Study population
The study sample was derived from participants in the baseline examination of the Multimodal Interventions to Delay Dementia and Disability in Rural China (MIND-China) study, as previously described in detail.[12] Briefly, MIND-China targeted local rural residents who were aged ≥60 years by the end of 2017 and living in the 52 villages of Yanlou Town, Yanggu County, western Shandong Province, China. Baseline multidisciplinary assessments were conducted in March–September 2018, during which 5765 participants were examined.[12]

In addition to testing the effectiveness of multimodal interventions to delay dementia and functional dependence in rural older adults, we intended to identify objective biomarkers for early detection of brain ageing and cognitive impairment (eg, mild cognitive impairment).[12] Retinal measurements assessed with SD-OCT hold great potential for the early diagnosis, prognosis and risk assessment of cognitive impairment.[4] The SD-OCT substudy was embedded within MIND-China and conducted at two local hospitals from March to June 2018 in Yanlou Town Hospital using Primus 200 OCT scanner (Carl Zeiss

Meditec, Germany) and from June 2019 to November 2020 in Southwest Lu Hospital using Spectralis OCT scanner (Software VV.1.10.2.0; Heidelberg Engineering, Heidelberg, Germany). The cluster (village)-randomised sampling approach was used for selecting participants in the SD-OCT substudy. In total, 1083 subjects (76.7% of all eligible persons who undertook baseline examinations of MIND-China in March–June 2018) from 16 villages in Yanlou Town Hospital and 306 subjects (33.5% of all eligible persons who undertook the MRI substudy of MIND-China in June 2019–November 2020) from 20 villages in Southwest Lu Hospital underwent the SD-OCT scans. We excluded 383 persons scanned with Primus 200 and 35 persons scanned with Spectralis due to the following reasons: suboptimal image quality (eg, signal strength less than six, segmentation and centration errors, and movement artefacts) (n=252 for Primus 200; n=21 for Spectralis), retinal disorders (eg, epiretinal membrane, macular oedema and macular hole) (n=125 for Primus 200; n=11 for Spectralis) and subjects with self-reported history of neurological disease (eg, multiple sclerosis) (n=5 for Primus 200; n=3 for Spectralis) and subjects aged <65 years (n=1 for Primus 200). Finally, 700 subjects examined with the Primus 200 OCT, and 271 subjects assessed with the Spectralis OCT were included in this analysis. Figure 1 shows the flow chart of the study participants in the SD-OCT substudy.

### Measurements and assessments of systemic factors
Data were collected via face-to-face interviews, clinical examination, laboratory test and ocular examinations following the standard procedures.[13] Arterial blood pressure was measured using an electronic manometer (Omron HEM-7127J; Omron Corporation, Kyoto, Japan) after at least a 5 min rest. After an overnight fast, peripheral blood samples were taken. Fasting serum glucose, total cholesterol, triglycerides, high-density lipoprotein cholesterol, low-density lipoprotein cholesterol were measured using an automatic Biochemical Analyzer (CS-600B, DIRUI Corporation, Changchun, China). Body mass index (BMI) was calculated as measured body weight (in kilograms) divided by body height (in metres) squared. We defined overweight as BMI 24–27.9 kg/m$^2$ and obesity as BMI≥28 kg/m$^2$ according to recommendations for Chinese adults.[14] Smoking status was categorised into current, former and never smoking. According to the frequency and quantity of alcohol consumption per week in the past 12 months, participants were divided into no or occasional drinkers and regular drinkers (at least once a week). Regular alcohol drinkers were further divided into light-to-moderate drinkers (<21 standard drinks/week for men and <14 standard drinks/week for women) and heavy drinkers (≥21 standard drinks/week for men, ≥14 standard drinks/week for women).[15] Physical inactivity was defined as fewer than 150 min per week spent in moderate or vigorous physical activities.[13] Hypertension was defined as systolic blood pressure ≥140 mmHg, or diastolic blood pressure≥90 mmHg, or current use of

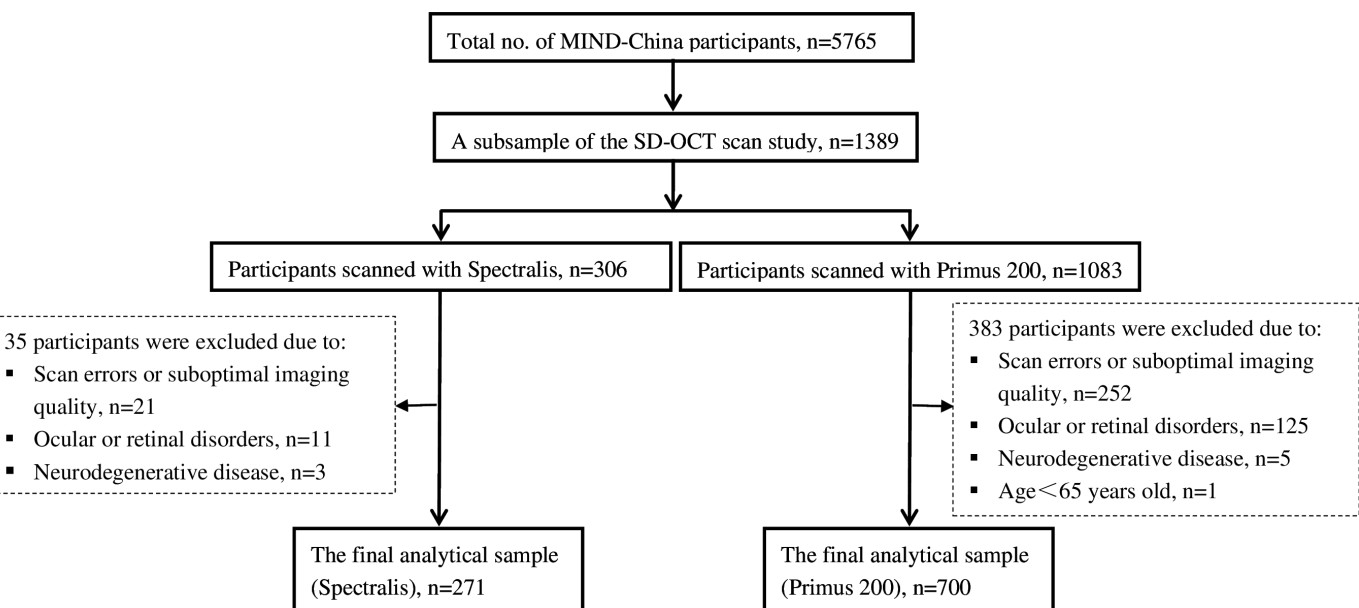

**Figure 1** Flow chart of study participants in the MIND-China SD-OCT substudy. MIND-China, the Multimodal Interventions to Delay Dementia and Disability in Rural China; SD-OCT, spectral-domain optical coherence tomography.

antihypertensive drugs.[16] Diabetes was defined as having a fasting plasma blood glucose level ≥7.0 mmol/L, or using hypoglycaemic drugs, or a self-reported history of diabetes.[17] Dyslipidaemia was defined as having a high-density lipoprotein level <1.0 mmol/L, or low-density lipoprotein level≥4.1 mmol/L, or triglycerides level ≥2.3 mmol/L, or total cholesterol ≥6.2 mmol/L, or using lipid-lowering drugs.[13] Cardiovascular disease (CVD) was ascertained from in-person interviews, clinical and neurological examinations, and ECG examination, which included ischaemic heart disease, heart failure, atrial fibrillation and stroke.[18]

## Measurements of ocular factors

All participants underwent presenting visual acuity assessment and split-lamp examination. In Southwest Lu Hospital, participants underwent additional eye examinations, including best corrected vision acuity, IOP, spherical equivalent, axial length and fundus imaging by trained ophthalmologist. Presenting and best corrected vision acuity were examined using a standard logarithmic visual acuity chart (Chinese edition, GB11533-2011) at a 5 m distance. Visual impairment was defined as presenting distance visual acuity <4.5 in the better eye.[19] Intraocular pressure was examined using non-contact tonometer (TX-20, Canon, Japan). Spherical equivalent was examined using computerised auto refractor (HDR-7000 system, Huvitz, Korea). Axial length was measured using A-scan ultrasound biometry (Compact TouchV.4.00, Quantel Medical, France). The anterior segment of the eye was examined using a handheld portable slit-lamp microscope (BM900, Haag-Streit Diagnostics, Switzerland). Both disc-cantered and macular-centred fundus images were taken through undilated pupils using a hand-held direct ophthalmoscope (Suzhou Liuliu Vision Technology., Suzhou, China).

## Imaging of OCT

The same procedure for SD-OCT examination was followed in the two hospitals. Following ocular examination, the SD-OCT scan was performed by experienced ophthalmic technicians. (1) The Primus 200 OCT has an acquisition rate of 12 000 A-scans per second, with an axial resolution of 5 µm in tissue and a transversal resolution of <20 µm using a super luminescence diode with an 870 nm bandwidth.[20] We scanned only one eye and the right eye was the first choice. The structure of macular retinal layers was analysed following the macular thickness analysis protocol based on 512×32 macular cube scan in a dark room without pupil dilation, 6 mm×6 mm in scan area, centred at the fovea. (2) The Spectralis OCT has an acquisition rate of 40 000 A-scans per second, with an axial resolution of 5 µm in tissue and a transversal resolution of <6 µm using a super luminescence diode with an 870 nm bandwidth. The built-in TruTrack active eye-tacking technology, which compensates for eye movement artefact, will improve the reproducibility of foveal centre selection during data acquisition.[21] Macular image was acquired using the perifoveal volumetric macular scan model (512 horizontal A scans per B scan, 49 horizontal B scans in 6 mm×6 mm area, no predetermined automatic real time) in a dark room without pupil dilation. The right eye was imaged first and then for the left eye.

Retinal thickness in the nine subfields defined by the Early Treatment Diabetic Retinopathy Study (ETDRS)[22] was acquired using the built-in software, as well as the general macular cube volume over the entire grid area. The standard retinal subfields are defined as central, inner superior, innerinferior, innernasal, innertemporal, outersuperior, outerinferior, outernasal and outertemporal. The central foveal, inner and outer subfields are bounded by the 1 mm, 3 mm and 6 mm diameter circle,

respectively. The whole retinal thickness was defined as the distance between internal limiting membrane (ILM) and retinal pigment epithelium boundary.

## Evaluation of SD-OCT images

Quality control measures included image quality score, an ILM indicator, and motion indicators, which helped identify cases with segmentation errors, blinks and eye motion artefacts.[20 23 24] ILM indicator was a measure of the minimum localised edge strength around the ILM boundary across the entire scan. It was useful for identifying segmentation errors and blink artefacts. The image quality of SD-OCT scans was assessed based on the range of the quality scores assigned by the respective instrument. For Primus 200 OCT, signal strength varies from 0 to 10, signal strength of 6 or more is considered adequate for image analysis.[20] For Spectralis OCT, the quality factor is a measure of signal-to-noise ratio expressed on a scale of 1–40 decibels. A quality reading of 25 decibels or more is considered adequate for image analysis.[21]

## Statistical analysis

IBM SPSS Statistics for Windows, V.25.0 (IBMA) was used for all statistical analysis. We presented mean (SD) for continuous variables and frequency (%) for categorical variables. We used one-way analysis of variance or t-test to compare continuous variables, and $\chi^2$ test for categorical variables. Multiple general linear regression analyses were performed to examine the association of macular thickness and volumes (dependent variables) with various risk factors (independent variables). Adjustments were made first for age and sex in model 1, and additional adjustments were made for all other factors (ie, education, smoking, alcohol consumption, CVD and visual impairment) in model 2. The collinearity of the independent variables in model 2 was checked by calculating variance inflation factors (VIFs). Variables with high VIFs were excluded from the model. Data were missing in 3 participants (0.3%) for hypertension and 7 (0.7%) for physical activity. Missing values in categorical variables were replaced with a dummy variable. For all analyses, a two-tailed p<0.05 was considered statistically significant.

## Patient and public involvement

None.

## RESULTS

### Characteristics of study participants

Characteristics of the study participants by two models of SD-OCT scanners were summarised in table 1. Compared with participants who were scanned with Primus 200 OCT, those scanned with Spectralis OCT were slightly younger (mean age 68.1 vs 70.4 years; p<0.001) and less likely to have CVD (25.1% vs 37.9%; p<0.001) but more likely to have visual impairment (7.7% vs 4.0%; p=0.017). The two groups did not differ significantly in the distributions of sex, education, smoking, alcohol consumption, physical activity, BMI, hypertension, diabetes and dyslipidaemia.

## Comparison of macular parameters assessed with Spectralis and Primus 200

Compared with macular parameters of participants examined with Primus 200 OCT scanner, those scanned with Spectralis OCT scanner had significantly higher macular thickness in all ETDRS subfields but a lower macular volume (p<0.001) (table 2). The association of macular parameters with type of scanner remained significant even after adjusting for age, sex and education (p<0.001).

## Correlates of macular parameters

As shown in figure 2, measurements of inner and outer macular thickness and macular volume were significantly decreased with increasing age in participants measured with both scanners. There was no significant difference with age in central macular thickness of both samples of participants scanned with Primus 200 OCT and Spectralis OCT. In the multiple linear regression analysis, the association of age with macular thickness and volume remained significant even after further adjusting for multiple potential confounding variables, but the association of age with average inner macular thickness became statistically nonsignificant in the sample of participants examined with Primus 200 OCT (table 3).

Female sex was significantly associated with lower central, average inner and outer macular thickness, and macular volume after adjusting for age in the sample of participants examined with both Spectralis OCT and Primus 200 OCT, and the association of female sex with average inner and outer macular thickness and macular volume in the sample of participants examined with Spectralis OCT and the association with central and average inner macular thickness in the sample of participants examined with Primus 200 OCT remained significant even after further adjusting for multiple potential confounding variables (table 3).

In the sample of participants scanned with Spectralis OCT, after adjusting for sex and age, middle school or above was significantly associated with higher central and average inner macular thickness, and the association with average inner macular thickness remained significant after further adjusting for multiple potential confounding variables (table 3). In addition, the presence of CVD was significantly associated with lower central macular thickness after adjusting for sex and age, but the association became statistically non-significant after further adjusting for multiple confounding factors. We also analysed the associations of ocular factors (ie, IOP, axial length and spherical equivalent) with macular parameters in the subsample of participants examined with Spectralis OCT. Higher spherical equivalent was significantly associated with lower average inner macular thickness after adjusting for sex and age, but the association became statistically non-significant after further adjusting for multiple confounding factors (table 3). No significant associations

**Table 1** Characteristics of the study participants in the analytical samples by OCT scanners

| Characteristics | Primus 200 (n=700) | Spectralis (n=271) | P value |
|---|---|---|---|
| Demographic factors | | | |
| Age, years, mean (SD) | 70.4 (4.2) | 68.1 (4.1) | <0.001 |
| Female sex, n (%) | 381 (54.4) | 153 (56.5) | 0.569 |
| Education, n (%) | | | 0.114 |
| Illiterate | 285 (40.7) | 91 (33.6) | |
| Elementary school | 297 (42.4) | 126 (46.5) | |
| Middle school or above | 118 (16.9) | 54 (19.9) | |
| Lifestyle factors | | | |
| Smoking status, n (%) | | | 0.110 |
| Never | 451 (64.4) | 192 (70.8) | |
| Former | 100 (14.3) | 27 (10.0) | |
| Current | 149 (21.3) | 52 (19.2) | |
| Alcohol consumption, n (%) | | | 0.657 |
| No or occasional | 495 (70.7) | 188 (69.4) | |
| Light-to-moderate | 169 (24.1) | 65 (24.0) | |
| Heavy | 36 (5.1) | 18 (6.6) | |
| Physical inactivity, n (%) | 447 (63.9) | 181 (66.8) | 0.391 |
| Clinical conditions | | | |
| BMI (kg/m$^2$), n (%) | | | 0.684 |
| Normal (<24) | 268 (38.3) | 112 (41.3) | |
| Overweight (24–27.9) | 291 (41.6) | 107 (39.5) | |
| Obesity (≥28) | 141 (20.1) | 52 (19.2) | |
| Hypertension, n (%) | 475 (67.9) | 186 (68.8) | 0.816 |
| Diabetes, n (%) | 94 (13.4) | 31 (11.4) | 0.406 |
| Dyslipidaemia, n (%) | 177 (25.3) | 60 (22.1) | 0.306 |
| Cardiovascular disease, n (%) | 265 (37.9) | 68 (25.1) | <0.001 |
| Visual impairment, n (%) | 28 (4.0) | 21 (7.7) | 0.017 |

In the analytical sample, data were missing in 3 participants (0.3%) for hypertension and 7 (0.7%) for physical activity. Missing values in categorical variables were replaced with a dummy variable.

P value is for the test of differences between people scanned with Spectralis OCT and those scanned with Primus 200 OCT.

OCT, optical coherence tomography; BMI, body mass index.

of IOP and axial length with macular measurements were found (data not shown).

## DISCUSSION

In this community-based SD-OCT scan study, we quantified macular thickness and volume among rural-dwelling Chinese older adults (age ≥60 years) using two models of SD-OCT scanners, compared macular parameters examined with Spectralis and Primus 200 OCT, and further explored demographics, lifestyle factors, clinical conditions and ocular factors associated with macular thickness and volume.

Previous studies have shown that macular parameters measured with different SD-OCT scanners vary substantially.[25] We found that, compared with Primus 200 OCT, Spectralis OCT scanner yielded higher macular thickness but a lower macular volume, which was consistent with reports from the literature.[25 26] Different software algorithms were used in different OCT scanners, and the segmentation software of Spectralis OCT delineates the retinal pigment epithelium in a deeper layer than Primus 200 OCT.[25] This could be the main reason for higher macular thickness assessed with Spectralis OCT. In addition, the differences in macular parameters might be partially attributed to the fact that the sample of participants scanned with Spectralis OCT was younger than those scanned with Primus 200 OCT in our study.

We found that the macula was thinnest in the central subfields, followed by outer subfields and thickest in the inner subfields. Nasal quadrant of the inner and outer subfields was thickest, followed by the superior quadrant. These findings are in line with the distribution of

**Table 2** Mean (SD) of macular thickness and volume in different macular subfields measured with Spectralis and Primus 200 OCT scanners

| Macular parameters | Primus 200* (n=700) | Spectralis* (n=271) | β coefficient (95% CI)† | P value |
|---|---|---|---|---|
| Macular thickness (µm) | | | | |
| Central macular subfield | 245.6 (26.0) | 263.7 (22.9) | 17.98 (14.43 to 21.53) | <0.001 |
| Inner macular subfields | | | | |
| Superior | 310.9 (23.3) | 331.7 (18.0) | 19.55 (16.43 to 22.67) | <0.001 |
| Inferior | 308.8 (25.6) | 327.6 (20.8) | 18.08 (14.60 to 21.55) | <0.001 |
| Nasal | 314.8 (20.2) | 334.3 (19.4) | 18.86 (16.07 to 21.66) | <0.001 |
| Temporal | 303.5 (19.5) | 319.9 (18.5) | 15.58 (12.88 to 18.29) | <0.001 |
| Average inner macular | 309.5 (18.6) | 328.4 (18.3) | 18.02 (15.42 to 20.61) | <0.001 |
| Outer macular subfields | | | | |
| Superior | 277.6 (20.9) | 293.1 (15.3) | 13.79 (10.99 to 16.59) | <0.001 |
| Inferior | 258.8 (23.9) | 283.9 (15.7) | 24.15 (20.99 to 27.32) | <0.001 |
| Nasal | 289.2 (19.3) | 310.6 (16.4) | 20.13 (17.48 to 22.79) | <0.001 |
| Temporal | 260.0 (16.4) | 281.0 (14.6) | 19.81 (17.54 to 22.08) | <0.001 |
| Average outer macular | 271.4 (15.3) | 292.2 (14.0) | 19.47 (17.35 to 21.60) | <0.001 |
| Macular volume ($mm^3$) | 9.8 (0.6) | 8.5 (0.4) | −1.34 (−1.42 to -1.26) | <0.001 |

*Data are mean (SD).
†β coefficient (95% CI) of macular parameters associated with scanner (Spectralis vs Primus 200), controlling for age, sex and education.
OCT, optical coherence tomography.

neurons and synapse in the macular region of retina and consistent with reports from previous population-based studies.[6 23 27–29] The thicknesses of central, average inner and average outer macula of population scanned with Primus 200 OCT were slightly thinner than the reports from the Singapore Epidemiology of Eye Diseases Study of Chinese population (age ≥40 years)[27 28] and the UK Biobank study (age ≥40 years).[23] However, macular thickness assessed with Spectralis OCT was thicker than that of the aforementioned reports. Differences in sociodemographic characteristics (eg, age and sex) and OCT devices might partly contribute to the variation of macular thickness across studies.[25]

In our study, older age was related to lower average inner and outer macular thickness and macular volume. There was a trend towards decreasing in central macular thickness with increasing age, but the association was not significant. These results were consistent with the reports from previous studies[23 27 28 30] and were also in line with physiological characteristics of the retina.[31 32] Histological studies show that nerve fibres and ganglion cells with higher density in the peripheral part of the retina seem more vulnerable to loss with ageing, while photoreceptors with higher density in the fovea are more resistant to age-related cell death.[31 32]

Previous studies have consistently shown that women have thinner macular thickness in all subfields and smaller macular volume than men.[6 23 27 29] Our data showed similar sex differences in the examined macular parameters. Sex differences in macular thickness and volume may be attributed to hormonal changes at menopause,

and this may explain why women were more likely than men to develop macular holes.[29]

We also found that achieving middle school or above was associated with higher macular thickness in the sample of participants scanned with Spectralis OCT, but not in those scanned with Primus 200 OCT. The differences in demographics of both samples (eg, mean age 68.1 for persons scanned with Spectralis vs 70.4 years for persons scanned with Primus 200; p<0.001) and the different sensitivity of the two scanners might partly contribute to the discrepant results. The UK Biobank study showed that higher education attainment was associated with higher thickness of macular retinal nerve fibre layer, ganglion cell complex and ganglion cell-inner plexiform layer, but the association between overall macular thickness and education attainment was not reported in this study.[33] Further large-scale population-based studies are needed to clarify the association of educational attainment with macular thickness.

Several population-based studies have investigated the associations of lifestyle factors and clinical conditions with macular measurements,[6 9 23 27–29] and some studies found that alcohol consumption, smoking,[23] high BMI,[6 23] high fasting blood glucose[6] or diabetes,[28] low levels of total cholesterol and triglyceride, and chronic kidney disease[28] were associated with lower macular measurements. Our data showed that CVD was associated with lower central macular thickness in the population scanned with Spectralis OCT. However, we did not find associations of common lifestyle factors and other clinical conditions with macular parameters. The exact

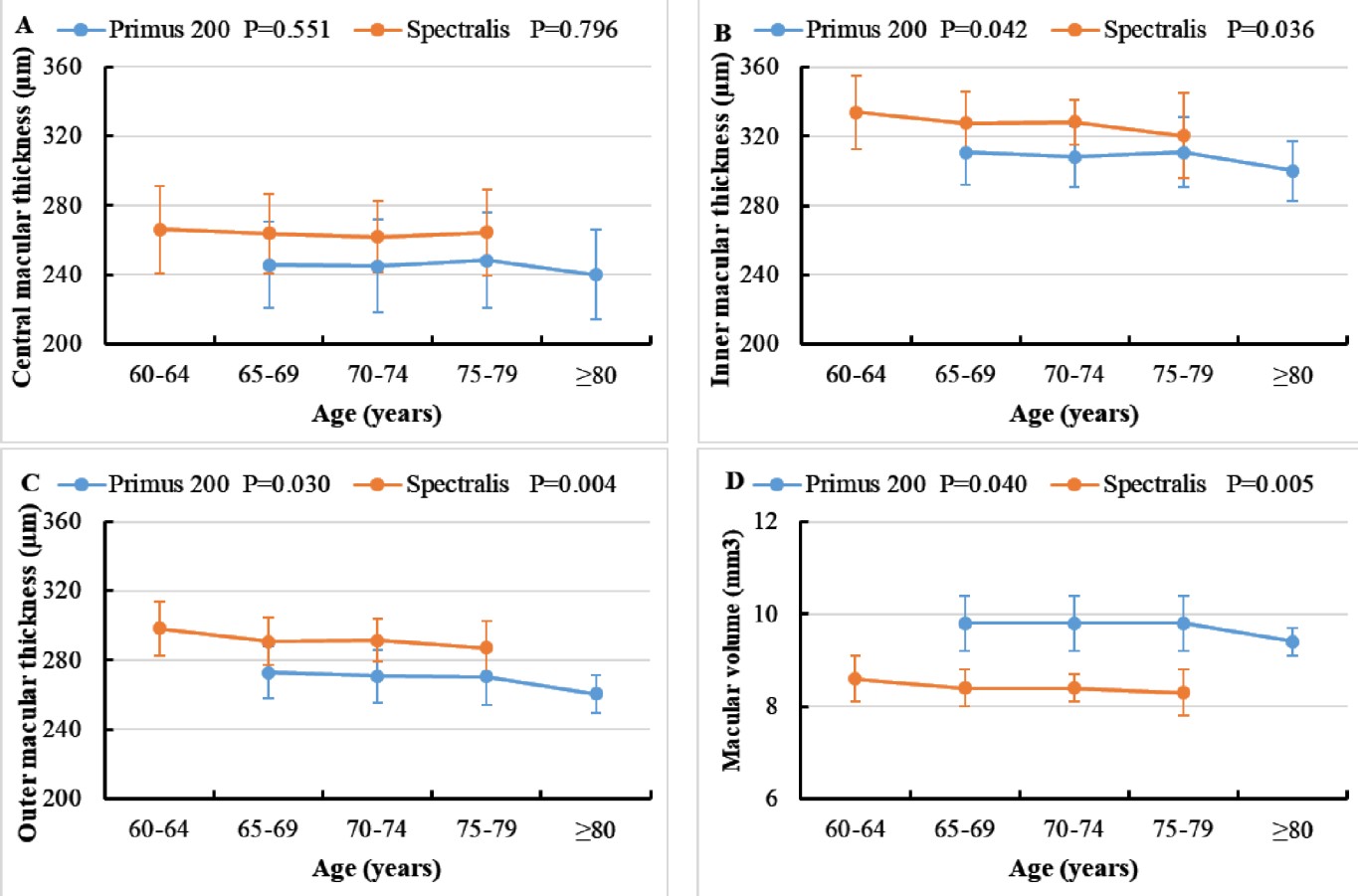

**Figure 2** Distributions of macular thicknesses and volume by age groups assessed with Spectralis and Primus 200 OCT scanners. Data are mean (SD). (A) Central macular thickness, (B) average inner macular thickness, (C) average outer macular thickness and (D) macular volume. P value was for the test of trend of macular parameters across age groups. OCT, optical coherence tomography.

reasons for the discrepant results across studies are not clear, but the differences in sociodemographic features of study population and some methodological issues across studies might play a part. Of note, our study participants were derived from a typical rural area in China, where older people had low socioeconomic status, the majority of them received no or very limited education, and they had distinct lifestyles from those of urban populations. Pathophysiologically, macular thickness could be affected by hypertension.[23 27 28] The retina undergoes various pathophysiological changes in patients with hypertension, such as vasoconstriction, vascular sclerosis and exudative changes. Vasoconstriction and vascular sclerosis may contribute to macular thinning, while exudative changes may induce abnormal macular thickening.[9 34] Therefore, it is important to evaluate ocular fundus changes before OCT examination. However, in our study, only participants examined with Spectralis OCT in Southwest Lu Hospital underwent fundus imaging evaluation. In addition, macular thickness is affected by diabetic retinopathy and its severity.[28 35] Unfortunately, diabetic retinopathy was not evaluated in our study. Further studies with careful assessment of diabetic retinopathy may help determine the relationships of macular measurements with diabetes

and complications. Regular physical activity may promote ocular health, but no previous studies have analysed the effect of physical activity on macular thickness. We found no association between physical inactivity and macular parameters among the rural-dwelling older adults.

Macular thickness and volume also varied with ocular factors. We found that lower spherical equivalent was associated with higher average inner macular thickness in the sample of participants scanned with Spectralis, which was consistent with reports from previous studies[23 36] and histological findings of retinal thinning in myopic eyes.[37 38] However, as there was no autorefractor in Yanlou Town Hospital, spherical equivalent was not examined for participants scanned with the Primus 200 device. Further large-scale population-based studies are warranted to clarify the association of spherical equivalent with macular thickness.

The strengths of our study include the population-based design, the use of SD-OCT rather than TD–OCT scanners and standardised assessment of wide range of systemic and ocular factors. Our study targeted older adults (age ≥60 years) who were living in the rural communities, whereas the Handan Eye Study was the only population-based OCT study in China that investigated

**Table 3** β coefficients (95% CI) of macular measurements associated with systemic and ocular factors from different models

| Factors | Central macular (μm) | | | | Average inner macular (μm) | | | | Average outer macular (μm) | | | | Volume (mm³) | | | |
|---|---|---|---|---|---|---|---|---|---|---|---|---|---|---|---|---|
| | Model 1* | P value | Model 2* | P value | Model 1* | P value | Model 2* | P value | Model 1* | P value | Model 2* | P value | Model 1* | P value | Model 2* | P value |
| **Spectralis OCT (n=271)** | | | | | | | | | | | | | | | | |
| Age (per 5 years) | −1.46 (−4.78 to 1.85) | 0.385 | −1.01 (−4.36 to 2.33) | 0.551 | −3.62 (−6.17 to −1.06) | 0.006 | −3.34 (−5.91 to −0.77) | 0.011 | −3.43 (−5.42 to −1.44) | 0.001 | −3.39 (−5.42 to −1.36) | 0.001 | −0.10 (−0.15 to −0.04) | 0.001 | −0.10 (−0.15 to −0.04) | 0.001 |
| Sex (female vs male) | −10.45 (−15.86 to −5.03) | <0.001 | −5.32 (−13.67 to 3.03) | 0.211 | −11.63 (−15.80 to −7.45) | <0.001 | −10.20 (−16.61 to −3.80) | 0.002 | −5.89 (−9.15 to −2.63) | <0.001 | −5.13 (−10.19 to −0.06) | 0.047 | −0.21 (−0.30 to −0.11) | <0.001 | −0.18 (−0.32 to −0.03) | 0.016 |
| **Education** | | | | | | | | | | | | | | | | |
| Illiterate | 0.00 (ref.) | | 0.00 (ref.) | | 0.00 (ref.) | | 0.00 (ref.) | | 0.00 (ref.) | | 0.00 (ref.) | | 0.00 (ref.) | | 0.00 (ref.) | |
| Elementary school | 3.10 (−3.22 to 9.42) | 0.335 | 2.96 (−3.44 to 9.35) | 0.363 | 3.70 (−1.16 to 8.55) | 0.135 | 3.81 (−1.09 to 8.72) | 0.127 | 2.12 (−1.70 to 5.94) | 0.275 | 2.35 (−1.52 to 6.23) | 0.233 | 0.07 (−0.04 to 0.18) | 0.202 | 0.08 (−0.03 to 0.19) | 0.174 |
| Middle school or above | 8.86 (0.27 to 17.44) | 0.043 | 8.24 (−0.50 to 16.98) | 0.065 | 8.49 (1.90 to 15.09) | 0.012 | 7.85 (1.14 to 14.55) | 0.022 | 3.76 (−1.43 to 8.94) | 0.155 | 3.61 (−1.69 to 8.91) | 0.181 | 0.14 (−0.01 to 0.28) | 0.066 | 0.13 (−0.02 to 0.28) | 0.089 |
| **Smoking** | | | | | | | | | | | | | | | | |
| Never | 0.00 (ref.) | | 0.00 (ref.) | | 0.00 (ref.) | | 0.00 (ref.) | | 0.00 (ref.) | | 0.00 (ref.) | | 0.00 (ref.) | | 0.00 (ref.) | |
| Former | −2.84 (−13.57 to 7.89) | 0.603 | −3.90 (−14.85 to 7.05) | 0.484 | −3.21 (−11.52 to 5.10) | 0.447 | −4.32 (−12.72 to 4.08) | 0.312 | 0.08 (−6.41 to 6.58) | 0.979 | 0.14 (−6.50 to 6.78) | 0.967 | −0.02 (−0.20 to 0.17) | 0.846 | −0.02 (−0.21 to 0.16) | 0.800 |
| Current | 4.93 (−3.78 to 13.65) | 0.266 | 3.37 (−5.76 to 12.50) | 0.468 | −0.16 (−6.91 to 6.58) | 0.962 | −1.31 (−8.32 to 5.69) | 0.712 | 0.71 (−4.56 to 5.98) | 0.792 | 0.58 (−4.96 to 6.12) | 0.837 | 0.02 (−0.13 to 0.17) | 0.801 | 0.01 (−0.15 to 0.16) | 0.917 |
| **Alcohol consumption** | | | | | | | | | | | | | | | | |
| No or occasional | 0.00 (ref.) | | 0.00 (ref.) | | 0.00 (ref.) | | 0.00 (ref.) | | 0.00 (ref.) | | 0.00 (ref.) | | 0.00 (ref.) | | 0.00 (ref.) | |
| Light to moderate | 3.39 (−4.29 to 11.06) | 0.386 | 1.74 (−6.17 to 9.65) | 0.665 | 0.36 (−5.57 to 6.29) | 0.904 | −0.69 (−6.77 to 5.38) | 0.822 | −1.73 (−6.35 to 2.89) | 0.462 | −2.53 (−7.33 to 2.27) | 0.300 | −0.03 (−0.16 to 0.10) | 0.624 | −0.06 (−0.19 to 0.08) | 0.401 |
| Heavy | 3.94 (−7.83 to 15.71) | 0.510 | 1.68 (−10.50 to 13.86) | 0.786 | 0.64 (−8.45 to 9.74) | 0.889 | 0.81 (−8.54 to 10.16) | 0.864 | 0.64 (−6.45 to 7.72) | 0.859 | 0.58 (−6.81 to 7.97) | 0.878 | 0.02 (−0.18 to 0.22) | 0.833 | 0.02 (−0.19 to 0.23) | 0.857 |
| CVD | −6.83 (−13.08 to −0.58) | 0.032 | −6.05 (−12.41 to 0.31) | 0.062 | −3.09 (−7.94 to 1.76) | 0.211 | −2.59 (−7.47 to 2.29) | 0.296 | −1.09 (−4.88 to 2.70) | 0.572 | −1.01 (−4.87 to 2.85) | 0.607 | −0.05 (−0.16 to 0.06) | 0.375 | −0.04 (−0.15 to 0.07) | 0.433 |
| Visual impairment | −1.50 (−11.61 to 8.62) | 0.771 | −0.03 (−10.26 to 10.20) | 0.995 | −4.79 (−12.58 to 2.99) | 0.226 | −4.67 (−12.51 to 3.18) | 0.243 | −1.13 (−7.22 to 4.96) | 0.715 | −1.07 (−7.28 to 5.14) | 0.734 | −0.05 (−0.23 to 0.12) | 0.541 | −0.05 (−0.23 to 0.13) | 0.572 |
| Spherical equivalent | −0.27 (−2.51 to 1.97) | 0.811 | −0.09 (−2.34 to 2.16) | 0.937 | −1.78 (−3.50 to −0.07) | 0.042 | −1.67 (−3.39 to 0.06) | 0.058 | −1.17 (−2.51 to 0.17) | 0.087 | −1.15 (−2.51 to 0.21) | 0.098 | −0.04 (−0.07 to 0.00) | 0.060 | −0.04 (−0.07 to 0.00) | 0.073 |
| **Primus 200 OCT (n=700)** | | | | | | | | | | | | | | | | |

Continued

**Table 3** Continued

| Factors | Central macular (μm) | | | | Average inner macular (μm) | | | | Average outer macular (μm) | | | | Volume (mm³) | | | |
|---|---|---|---|---|---|---|---|---|---|---|---|---|---|---|---|---|
| | Model 1* | P value | Model 2* | P value | Model 1* | P value | Model 2* | P value | Model 1* | P value | Model 2* | P value | Model 1* | P value | Model 2* | P value |
| Age (per 5 years) | −0.08 (−2.31 to 2.15) | 0.941 | 0.10 (−2.16 to 2.36) | 0.931 | −1.47 (−3.07 to 0.12) | 0.070 | −1.36 (−2.98 to 0.26) | 0.101 | −2.34 (−3.67 to −1.01) | 0.001 | −2.32 (−3.67 to −0.97) | 0.001 | −0.07 (−0.12 to −0.02) | 0.008 | −0.06 (−0.12 to −0.01) | 0.013 |
| Sex (female vs male) | −13.31 (−17.07 to −9.55) | <0.001 | −11.06 (−18.10 to −4.02) | 0.002 | −8.72 (−11.41 to −6.02) | <0.001 | −7.47 (−12.51 to −2.43) | 0.004 | −4.32 (−6.57 to −2.08) | <0.001 | −3.33 (−7.54 to 0.88) | 0.121 | −0.15 (−0.24 to −0.07) | 0.001 | −0.12 (−0.28 to 0.04) | 0.137 |
| Education | | | | | | | | | | | | | | | | |
| Illiterate | 0.00 (ref.) | | 0.00 (ref.) | | 0.00 (ref.) | | 0.00 (ref.) | | 0.00 (ref.) | | 0.00 (ref.) | | 0.00 (ref.) | | 0.00 (ref.) | |
| Elementary school | −1.90 (−6.39 to 2.59) | 0.406 | −2.02 (−6.55 to 2.50) | 0.380 | −0.35 (−3.57 to 2.88) | 0.833 | −0.58 (−3.82 to 2.66) | 0.724 | 0.43 (−2.25 to 3.12) | 0.751 | 0.34 (−2.37 to 3.04) | 0.807 | 0.00 (−0.10 to 0.10) | 0.988 | −0.01 (−0.11 to 0.10) | 0.905 |
| Middle school or above | 2.93 (−3.32 to 9.17) | 0.358 | 2.94 (−3.34 to 9.22) | 0.358 | 0.88 (−3.59 to 5.36) | 0.699 | 0.62 (−3.87 to 5.12) | 0.785 | 2.05 (−1.68 to 5.78) | 0.282 | 1.92 (−1.83 to 5.67) | 0.315 | 0.09 (−0.05 to 0.23) | 0.222 | 0.08 (−0.06 to 0.22) | 0.284 |
| Smoking | | | | | | | | | | | | | | | | |
| Never | 0.00 (ref.) | | 0.00 (ref.) | | 0.00 (ref.) | | 0.00 (ref.) | | 0.00 (ref.) | | 0.00 (ref.) | | 0.00 (ref.) | | 0.00 (ref.) | |
| Former | 1.50 (−6.16 to 9.15) | 0.701 | 2.24 (−5.49 to 9.98) | 0.570 | 2.90 (−2.58 to 8.37) | 0.299 | 2.93 (−2.61 to 8.47) | 0.300 | 1.58 (−2.99 to 6.14) | 0.497 | 1.83 (−2.79 to 6.46) | 0.437 | 0.06 (−0.11 to 0.24) | 0.464 | 0.07 (−0.11 to 0.24) | 0.449 |
| Current | 3.31 (−3.79 to 10.40) | 0.360 | 4.32 (−2.97 to 11.60) | 0.245 | −0.31 (−5.38 to 4.76) | 0.905 | −0.11 (−5.33 to 5.11) | 0.966 | −1.22 (−5.44 to 3.01) | 0.572 | −1.14 (−5.50 to 3.21) | 0.607 | −0.08 (−0.24 to 0.08) | 0.343 | −0.08 (−0.24 to 0.09) | 0.354 |
| Alcohol consumption | | | | | | | | | | | | | | | | |
| No or occasional | 0.00 (ref.) | | 0.00 (ref.) | | 0.00 (ref.) | | 0.00 (ref.) | | 0.00 (ref.) | | 0.00 (ref.) | | 0.00 (ref.) | | 0.00 (ref.) | |
| Light to moderate | −2.34 (−7.52 to 2.85) | 0.376 | −2.71 (−8.00 to 2.57) | 0.314 | 0.55 (−3.17 to 4.26) | 0.773 | 0.64 (−3.15 to 4.43) | 0.740 | 0.53 (−2.57 to 3.63) | 0.737 | 0.66 (−2.50 to 3.82) | 0.681 | 0.03 (−0.09 to 0.15) | 0.622 | 0.04 (−0.08 to 0.16) | 0.489 |
| Heavy | 1.52 (−7.65 to 10.68) | 0.745 | 0.59 (−8.74 to 9.93) | 0.901 | −1.41 (−7.97 to 5.15) | 0.673 | −1.12 (−7.81 to 5.56) | 0.742 | −0.28 (−5.75 to 5.19) | 0.920 | 0.03 (−5.55 to 5.61) | 0.991 | 0.01 (−0.20 to 0.21) | 0.941 | 0.03 (−0.18 to 0.24) | 0.755 |
| CVD | −1.40 (−5.27 to 2.47) | 0.477 | −1.51 (−5.41 to 2.39) | 0.448 | −0.35 (−3.12 to 2.42) | 0.802 | −0.52 (−3.31 to 2.27) | 0.715 | −1.27 (−3.58 to 1.03) | 0.279 | −1.39 (−3.72 to 0.94) | 0.243 | −0.03 (−0.12 to 0.06) | 0.504 | −0.03 (−0.12 to 0.05) | 0.444 |
| Visual impairment | −4.47 (−14.09 to 5.15) | 0.362 | −4.67 (−14.32 to 4.99) | 0.343 | −6.39 (−13.26 to 0.48) | 0.068 | −6.19 (−13.10 to 0.73) | 0.079 | −0.54 (−6.28 to 5.21) | 0.854 | −0.17 (−5.94 to 5.61) | 0.955 | −0.14 (−0.35 to 0.08) | 0.221 | −0.12 (−0.34 to 0.10) | 0.272 |

*β coefficients (95% CI) were adjusted for age and sex in model 1; and additionally adjusted for all other factors in the table in model 2.
CVD, cardiovascular disease; OCT, optical coherence tomography.

macular structures mainly among young and middle-aged urban residents (age ≥30 years, mean age 46.4 years).[6] Furthermore, we examined a broader range of demographic, lifestyle, clinical and ocular factors associated with macular parameters that were not reported in the previous study.[6] However, our study has limitations. First, ocular factors were acquired only in a small sample, thus, the study has limited power to analyse associations of these factors with macular parameters. Second, the cross-sectional nature of our study does not allow us to make any causal inference for the observed associations between factors studied and macular parameters, and the observed cross-sectional associations might be subject to selective survival bias that usually leads to underestimation of the true associations. Third, we were not able to control for the potential confounding effects of some ocular diseases (eg, age-related macular degeneration and diabetic retinopathy) due to lack of assessments of these disorders, and residual confounding might also exist due to imperfect assessments of some confounding factors. Finally, our study samples were relatively younger, more educated and more likely to smoke and have visual impairment compared with the total MIND-China sample, which should be kept in mind when generalising our findings to other populations.

## CONCLUSION

This population-based SD-OCT study of older adults who were free of ocular disorders quantified the macular thicknesses and volumes. We found that older age and female sex were associated with thinner macular thickness. Furthermore, Spectralis OCT produced higher macular thickness but a lower macular volume than Primus 200 OCT. These data may help interpret macular parameters in the diagnosis and management of retinal disorders in older adults. In addition, our study provides detailed description of baseline macular measurements in a subsample of the community-based MIND-China study, which is highly relevant for future follow-up studies that investigate the longitudinal relationships of macular parameters with measures of brain ageing and functional phenotypes.

**Author affiliations**
[1]Department of Neurology, Shandong Provincial Hospital Affiliated to Shandong First Medical University, Jinan, Shandong, China
[2]Department of Neurology, Shandong Provincial Hospital, Cheeloo College of Medicine, Shandong University, Jinan, Shandong, People's Republic of China
[3]Shandong Provincial Clinical Research Center for Geriatric Neurological Diseases, Jinan, Shandong, P. R. China
[4]Department of Ophthalmology, Shandong Provincial Hospital Affiliated to Shandong First Medical University, Jinan, Shandong, China
[5]Department of Ophthalmology, Shandong Provincial Hospital, Cheeloo College of Medicine, Shandong University, Jinan, Shandong, People's Republic of China
[6]Department of Neurology, Shandong Provincial ENT Hospital, Jinan, Shandong, China
[7]Department of Neurobiology, Aging Research Center and Center for Alzheimer Research, Care Sciences and Society, Karolinska Institutet-Stockholm University, Stockholm, Sweden

**Acknowledgements** We would like to thank all participants of the MIND-China project as well as our research staff at Yanlou Town Hospital, Southwest Lu Hospital, and the Department of Neurology, Shandong Provincial Hospital who were involved in the data collection and management.

**Contributors** QZ, LC, YW, YDu and CQ contributed to the study concept and design. QZ, CoZ, ZX and CJ contributed to the acquisition of data; QZ, YDong and JF contributed to the assessment of data. WZ and CoZ contributed to the assessment of OCT images; QZ and KL contributed to the data analysis. QZ, WZ and ChZ contributed to interpretation of the results; QZ contributed to the drafting of the manuscript. YDu and CQ contributed to the study supervision. All authors contributed to the critical revision of the manuscript for important intellectual content, article and approved the submitted version. QZ is responsible for the overall content as guarantor.

**Funding** This work was supported in part by grants from the National Key R&D Program of China (grant no.: 2017YFC1310100), the National Nature Science Foundation of China (grants no.: 82171175, 81861138008 and 81772448), the Academic Promotion Program of Shandong First Medical University, the Taishan Scholar Program of Shandong Province, Shandong Province postdoctoral innovation research program and the Jinan Science and technology development plan (grant no. 201805076), China. CQ received grants from the Swedish Research Council (VR, grants no.: 2017-05819 and 2020-01574) and the Swedish Foundation for International Cooperation in Research and Higher Education (STINT, grant no.: CH2019-8320) and Karolinska Institutet, Stockholm, Sweden.

**Disclaimer** The funding agency had no role in the study design, data collection and analysis, the writing of this manuscript, and in the decision to submit the work for publication.

**Competing interests** None declared.

**Patient and public involvement** Patients and/or the public were involved in the design, or conduct, or reporting, or dissemination plans of this research. Refer to the Methods section for further details.

**Patient consent for publication** Not applicable.

**Ethics approval** The protocol of the MIND-China study was reviewed and approved by the Ethics Committee at Shandong Provincial Hospital, Jinan, Shandong, China. Written informed consent was obtained from all participants, or in case of illiterate participants, from legally authorised representatives.

**Provenance and peer review** Not commissioned; externally peer reviewed.

**Data availability statement** Data are available on reasonable request.

**ORCID iDs**
Qinghua Zhang http://orcid.org/0000-0001-6799-9902
YiFeng Du http://orcid.org/0000-0002-4672-6654

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
