## [Reviewer comments · BMJ Open]

ARTICLE DETAILS

TITLE (PROVISIONAL)	Quantitative assessments of retinal macular structure among rural-dwelling older adults in China: a population-based, cross-sectional, optical coherence tomography study
AUTHORS	zhang, qinghua; Zhang, Cong; Wang, Yongxiang; Cong, Lin; Liu, Keke; Xu, Zhe; Jiang, Chunyan; Zhou, Weiyan; Zhang, Chunxiao; Dong, Yi; Feng, Jianli; Qiu, Chengxuan; Du, YiFeng

VERSION 1 – REVIEW

REVIEWER	Matoba, Ryo Okayama University Graduate School of Medicine Dentistry and Pharmaceutical Sciences
REVIEW RETURNED	16-Oct-2023

GENERAL COMMENTS	This is an important study that examines basic information about macular parameters assessed using the two SD-OCT scanners in rural-dwelling older adults in China. Overall, the study is well written, but the following points need to be corrected. Major 1) For factors such as achieving middle school and above and having CVD that were significantly different only in one model of OCT, it is questionable to conclude that "macular parameters were also affected by educational attainment, CVD, and spherical equivalent"(line343–344). What is your rationale for excluding the Primus results, which have a larger sample size? Please add that reason and revise your conclusion to what logically follows from the results. Minor 2) line 214– (throughout the Results section) and Table 3 Instead of "P<0.05" and "P<0.01", please provide specific P values to enhance the clarity and precision of the reported statistical significance. 3) Figure 2 Please include an explanation of the statistical differences observed to better interpret the findings.
---

REVIEWER	Antropoli, Alessio Università Vita Salute San Raffaele, Ophthalmology
REVIEW RETURNED	29-Oct-2023

GENERAL COMMENTS	This article is well-written, with detailed methods and a sizable cohort. While it doesn't introduce many new insights into existing knowledge, there are some noteworthy points that could be
--

	highlighted more. For instance, the authors performed the same analyses with two different devices and found that the correlations between OCT parameters and various collected data, such as CVD, spherical equivalent, education level, and vision impairment, were only statistically significant with the Spectralis device. It would be helpful for the authors to explain why these correlations were only observed with the Spectralis device. Additionally, it's important to understand what the authors believe is the primary cause of visual impairment in their cohorts. Since they excluded retinal diseases, it's likely that cataracts were a common cause. Were cataracts investigated or reported in the study? Clarifying this aspect would enhance the completeness of the research. Lastly, I will further clarify in the discussion the importance of their results and how and why they serve the purpose of the MIND-China study.
--	---

VERSION 1 – AUTHOR RESPONSE

Reviewer: 1

Dr. Ryo Matoba, Okayama University Graduate School of Medicine Dentistry and Pharmaceutical Sciences

Comments to the Author:

This is an important study that examines basic information about macular parameters assessed using the two SD-OCT scanners in rural-dwelling older adults in China. Overall, the study is well written, but the following points need to be corrected.

Response: We are grateful to the reviewer for the overall positive comments, along with kind suggestions for helping us to improve our work.

Major

1) For factors such as achieving middle school and above and having CVD that were significantly different only in one model of OCT, it is questionable to conclude that "macular parameters were also affected by educational attainment, CVD, and spherical equivalent"(line343–344). What is your rationale for excluding the Primus results, which have a larger sample size? Please add that reason and revise your conclusion to what logically follows from the results.

Response: We thank the reviewer for the very important comments that we agree with. We have now rephrased our conclusions and delete the sentence "macular parameters were also affected by educational attainment, CVD, and spherical equivalent". We also briefly discussed about some discrepant results from two models of devices (see page 12, lines 294-297, and pages 13-14, lines 331-334).

Minor

2) line 214– (throughout the Results section) and Table 3

Instead of "P<0.05" and "P<0.01", please provide specific P values to enhance the clarity and precision of the reported statistical significance.

Response: We thank the reviewer for careful review and valuable comments. We now reported the exact P values in Table 3 (see pages 24-25, Table 3) and further provide the exact P values in the results of the revised version of the manuscript. (see page 9, lines 214-215).

3) Figure 2

Please include an explanation of the statistical differences observed to better interpret the findings.

Response: We thank the reviewer for the kind suggestion. We now added the "Note: P was for the test of trend of macular parameters across age groups." to the legend for Figure 2 and added the P

value in Figure 2 in response to the Reviewer's comments (see page 27, Figure 2, and page 21, line 515).

Reviewer: 2

Dr. Alessio Antropoli, Università Vita Salute San Raffaele

Comments to the Author:

This article is well-written, with detailed methods and a sizable cohort. While it doesn't introduce many new insights into existing knowledge, there are some noteworthy points that could be highlighted more. For instance, the authors performed the same analyses with two different devices and found that the correlations between OCT parameters and various collected data, such as CVD, spherical equivalent, education level, and vision impairment, were only statistically significant with the Spectralis device. It would be helpful for the authors to explain why these correlations were only observed with the Spectralis device.

Response: We are grateful to the reviewer for the overall positive comments, along with kind suggestions for helping to improve our work.

(1) Indeed, previous population-based SD-OCT studies have shown that spherical equivalent was associated with thinner macular thickness¹, which may correlate with axial elongation and loss of photoreceptor packing density in myopic eyes. However, as there was no autorefractor in Yanlou Town Hospital, spherical equivalent was not examined for participants scanned with the Primus 200 device. We have now briefly explained this in the text (see pages 13-14, lines 331-334).

(2) To our knowledge, very few studies have assessed the association of education and macular thickness², and the underlying mechanism was unclear. The differences in demographics of both sample (e.g., mean age 68.1 for persons scanned with Spectralis vs. 70.4 years for persons scanned with Primus 200; $P < 0.001$) and the different sensitivity of the two scanners might partly contribute to the discrepant results. We now revised the discussion about the association of education and macular thickness (see page 12, lines 294-297).

(3) Previous case-control studies showed thinner retina thickness in participants with heart failure and stroke^{3 4}, which may partially attribute to decrease in vessel density and regressive neuro-axonal damage. Our study showed that the presence of CVD was associated with thinner central macular thickness, but this association become statistically non-significant in the fully-adjusted model. Further population-based studies in diverse populations are needed to clarify the relationships of CVD with macular thickness.

Additionally, it's important to understand what the authors believe is the primary cause of visual impairment in their cohorts. Since they excluded retinal diseases, it's likely that cataracts were a common cause. Were cataracts investigated or reported in the study? Clarifying this aspect would enhance the completeness of the research.

Response: We appreciate the reviewer for valuable comments. We agree with the reviewer that cataracts were a common cause of visual impairment. We have excluded participants with low signal strength in the inclusion criteria of study participation. Therefore, we did not analyze the impact of cataract on macular parameters in our study.

Lastly, I will further clarify in the discussion the importance of their results and how and why they serve the purpose of the MIND-China study.

Response: We thank the reviewer for the very important comments. Our study engaged older adults who were living in rural communities in China and had received no or very limited school education and had relatively low socioeconomic status, and this sociodemographic group has been substantially under studied in the literature as we highlighted in the introduction. In addition, we studied a wider range of factors (e.g., demographic, lifestyle, and clinical and retinal factors) associated with retinal parameters in the current literature. Further, we compared retinal parameters measured with two most commonly used models of devices, which provided important information for studies using these two different models. We believe that these results from our study could contribute to the current literature.

One of the aims of the MIND-China study was to identify objective biomarkers for early detection of brain aging and cognitive impairment (e.g., MCI) and physical dysfunction (in addition to testing the effectiveness of multimodal interventions to delay dementia and functional dependence in rural older adults, as we previously described)⁵. Based on the current literature, we hypothesized that retinal measurements/biomarkers assessed with SD-OCT hold great potential for the early diagnosis, prognosis, and risk assessment of dementia/AD and MCI. We believe that the current descriptive study (e.g., methodological description and distribution of retinal parameters and related factors) will

be very important for future research (including follow-up studies) regarding early diagnosis and prognosis of MCI, and their associations with brain aging markers (e.g., plasma and MRI biomarkers for brain aging and AD measured in MIND-China. See Dong Y, et al. J Alzheimer's Dis 6) in rural residents. We have now briefly highlighted this in the manuscript (see page 5, lines 94-98).

References

1. von Hanno T, Lade AC, Mathiesen EB, et al. Macular thickness in healthy eyes of adults (N = 4508) and relation to sex, age and refraction: the Tromsø Eye Study (2007-2008). *Acta ophthalmologica* 2017;95(3):262-69. doi: 10.1111/aos.13337
2. Khawaja AP, Chua S, Hysi PG, et al. Comparison of Associations with Different Macular Inner Retinal Thickness Parameters in a Large Cohort: The UK Biobank. *Ophthalmology* 2020;127(1):62-71. doi: 10.1016/j.ophtha.2019.08.015
3. Ye C, Kwapong WR, Tao W, et al. Characterization of Macular Structural and Microvascular Changes in Thalamic Infarction Patients: A Swept-Source Optical Coherence Tomography-Angiography Study. *Brain Sci* 2022;12(5) doi: 10.3390/brainsci12050518
4. Khalilipour E, Mahdizad Z, Molazadeh N, et al. Microvascular and structural analysis of the retina and choroid in heart failure patients with reduced ejection fraction. *Sci Rep* 2023;13(1):5467. doi: 10.1038/s41598-023-32751-w
5. Wang Y, Han X, Zhang X, et al. Health status and risk profiles for brain aging of rural-dwelling older adults: Data from the interdisciplinary baseline assessments in MIND-China. *Alzheimers Dement (N Y)* 2022;8(1):e12254. doi: 10.1002/trc2.12254
6. Dong Y, Hou T, Li Y, et al. Plasma Amyloid-β, Total Tau, and Neurofilament Light Chain Across the Alzheimer's Disease Clinical Spectrum: A Population-Based Study. *J Alzheimers Dis* 2023 doi: 10.3233/JAD-230932

VERSION 2 – REVIEW

REVIEWER	Matoba, Ryo Okayama University Graduate School of Medicine Dentistry and Pharmaceutical Sciences
REVIEW RETURNED	09-Jan-2024
GENERAL COMMENTS	The authors have satisfyingly answered all my questions.
REVIEWER	Antropoli, Alessio Università Vita Salute San Raffaele, Ophthalmology
REVIEW RETURNED	16-Jan-2024
GENERAL COMMENTS	The authors have successfully addressed all the raised points, and I commend their work, which has definitely improved.